# Inertial and viscous flywheel sensing of nanoparticles

Georgios Katsikis [1,6], Jesse F. Collis [2,6], Scott M. Knudsen[1], Vincent Agache[1,3], John E. Sader [2✉] & Scott R. Manalis [1,4,5✉]

Rotational dynamics often challenge physical intuition while enabling unique realizations, from the rotor of a gyroscope that maintains its orientation regardless of the outer gimbals, to a tennis racket that rotates around its handle when tossed face-up in the air. In the context of inertial sensing, which can measure mass with atomic precision, rotational dynamics are normally considered a complication hindering measurement interpretation. Here, we exploit the rotational dynamics of a microfluidic device to develop a modality in inertial sensing. Combining theory with experiments, we show that this modality measures the volume of a rigid particle while normally being insensitive to its density. Paradoxically, particle density only emerges when fluid viscosity becomes dominant over inertia. We explain this paradox via a viscosity-driven, hydrodynamic coupling between the fluid and the particle that activates the rotational inertia of the particle, converting it into a 'viscous flywheel'. This modality now enables the simultaneous measurement of particle volume and mass in fluid, using a single, high-throughput measurement.

[1] Koch Institute for Integrative Cancer Research, Massachusetts Institute of Technology, Cambridge, MA, USA. [2] ARC Centre of Excellence in Exciton Science, School of Mathematics and Statistics, The University of Melbourne, VIC, Australia. [3] Université Grenoble Alpes, CEA, LETI, Grenoble, France. [4] Department of Biological Engineering, Massachusetts Institute of Technology, Cambridge, MA, USA. [5] Department of Mechanical Engineering, Massachusetts Institute of Technology, Cambridge, MA, USA. [6] These authors contributed equally: Georgios Katsikis, Jesse F. Collis. ✉email: jsader@unimelb.edu.au; srm@mit.edu

D rawing from model paradigms in classical physics such as mass-spring systems to everyday objects such as guitar strings[1], tuning forks[2], and bridge structures[3], the natural or resonant frequency of a given object is intuitively associated with its mass and stiffness. Generally, the heavier and softer the object, the lower its natural frequency. In inertial sensing, thin plates or long cantilevers, either hollow or rigid, are driven to oscillate at their resonant frequency[4–7]. When a particle traverses inside or lands upon the surface of the sensor[8–12], provided a local displacement exists, the particle changes the resonant frequency of the sensor in proportion to its mass[7,13]. Within this standard framework, rotational dynamics of the sensor are either ignored, or considered to be an erroneous or complicating factor in the measurement[14].

Here, we exploit rotational dynamics in inertial sensing to enable a measurement modality for rigid particles. We utilize the resonant frequency change induced when a particle is suspended in the fluid between microchannel walls that exhibit oscillatory rotation (Fig. 1a). We experimentally realize this motion using cantilevers in the form of suspended micro- and nanochannel resonators (SMRs/SNRs)[15,16]. The cantilevers vibrate in their second resonant flexural mode, where local rotation, without displacement, occurs at the vibrational nodes.

## Results

**Flow between oscillatory rotating walls**. Actuating the cantilever with no particle present at a vibrational node generates a flow within the microchannel, termed the base flow[17]. This flow depends on the viscous penetration depth, $\delta$, relative to the flow channel height, $H$, and is quantified by the oscillatory Reynolds number, $\beta = (H/\delta)^2$; which also specifies the ratio of inertial-to-viscous forces in the fluid. The viscous penetration depth, $\delta = \sqrt{\mu/(\rho\omega)}$, is the distance from a solid boundary—here, the rotating walls—into the fluid where viscous effects are important. The fluid has shear viscosity, $\mu$, and mass density, $\rho$; $\omega = 2\pi f$ is the angular frequency of the wall motion, which here is a resonant frequency of the cantilever. We assumed the channel is centered on the neutral axis of the cantilever; the effect of off-axis placement on our new modality is negligible (Supplementary Note 2).

To gain insight into the base flow, we explored the limits of low ($\beta \ll 1$) and high ($\beta \gg 1$) fluid inertia. For low inertia ($\beta \ll 1$), viscous effects dominate throughout the channel; the near-total absence of inertia in the fluid mostly occludes motion relative to the walls. Thus, the rotating walls generate a base flow that is primarily a rigid-body rotation (Fig. 1b, i). A finite, but small, amount of inertia in the fluid causes the fluid to slightly lag this primary base flow, producing a secondary shear flow that exhibits a nonlinear profile across the height of the channel (Fig. 1b, ii). This secondary flow can be decomposed into its fundamental rotational and extensional components (Supplementary Fig. 1a, ii-v). The superposition of the primary and secondary base flows gives the complete base flow for $\beta \ll 1$. For high inertia ($\beta \gg 1$), the fluid flow throughout the channel is predominantly inviscid. This inviscid flow is driven only by the normal component of the motion of the rotating walls, which periodically pushes and pulls on the fluid, producing an extensional, irrotational flow (Fig. 1b, iii).

**Theory for interaction between particle and rotating walls**. When a rigid particle is placed in the oscillatory base flow (Fig. 1b), a disturbance flow is generated (Fig. 1c, Supplementary Figs. 1b, 2) that produces a measurable signal. The two flows altogether satisfy the no-penetration and no-slip conditions at the particle's surface. This disturbance flow changes the hydrodynamic stress in the fluid which in turn applies an oscillatory

torque to the cantilever centered at the particle's longitudinal position. This torque modifies the cantilever's resonant frequency ($\Delta f_{\rm rot}$).

The shift $\Delta f_{\rm rot}$ is exclusively driven by the cantilever's local rotation and generally co-exists with a well-characterized frequency change $\Delta f_{\rm disp}$ due to the cantilever's translational displacement; this signal is typically used to measure the buoyant mass of a particle[4–8,13]. We developed a theory for $\Delta f_{\rm rot}$, for a particle of dimensional radius, $R = a/H$ where $a$ is its dimensional radius, and dimensionless mass density, $\gamma = \rho_{\rm p}/\rho$ where $\rho_{\rm p}$ is the density of the particle (Supplementary Note 1):

$$\Delta f_{\rm rot} = -f\alpha_{\rm v}\left(\beta\,|\,\gamma, R, z\right)\mathcal{V}\left(\frac{dW}{dx}\right)^2, \qquad (1)$$

where $f$ is the resonant frequency when no particle is present, $\mathcal{V} = \rho V^{5/3}/\left(2\left[6\pi^2\right]^{1/3} m_{\rm eff} L^2\right)$ is a dimensionless particle volume factor with particle volume $V$, $z$ is the vertical position of the particle within the channel scaled by the channel height, $H$, $L$ is the cantilever length, $m_{\rm eff}$ is the effective mass of the cantilever and $W(x)$ is the displacement mode shape of the cantilever along its length $x$, which is scaled by $L$. The viscous enhancement factor, $\alpha_{\rm v}$, defines the relative contribution of fluid viscosity to the frequency shift; $\alpha_{\rm v} \to 1$ for inviscid flow (Fig. 2, $\beta \gg 1$). The disturbance flow in this inviscid limit is identical to the flow derived from the scattered acoustic field at large wavelength, previously studied computationally[18].

While $\Delta f_{\rm rot}$ depends on both $\mathcal{V}$ and $\alpha_{\rm v}$, information on the rich variety of flow behavior is contained within $\alpha_{\rm v}$; $\mathcal{V}$ depends only on the particle's volume and properties of the cantilever. Specifically, $\alpha_{\rm v}$ exhibits a strong non-monotonic dependence on $\beta$ (Fig. 2), and a conditional dependence—with respect to $\beta$—on particle density, $\gamma$ (Fig. 2), radius, $R$ (Supplementary Fig. 3) and $z$-position (Supplementary Fig. 4, Supplementary Note 1).

**Understanding the limits of low and high inertia**. The general case for arbitrary $\beta$ is complicated, hence we achieved understanding by again exploring the limits of low ($\beta \ll 1$) and high ($\beta \gg 1$) inertia, for particles placed at the channel center ($z = 0$).

For low inertia ($\beta \ll 1$), the particle experiences the primary base flow of rigid-body rotation (Fig. 1b, i). Although fluid inertia is negligible, the particle's inertia causes it to rotate relative to the base flow, when particle density differs from that of the fluid ($\gamma \neq 1$). This generates a rotational disturbance flow (Fig. 1c, i, Supplementary Movie 1). Notably, in the limit $\beta \to 0$, the viscous enhancement factor, $\alpha_{\rm v}$, is constant and determined only by $\gamma$ (Fig. 2, $\beta \ll 1$). While a disturbance flow due to the (secondary) shear flow (Fig. 1b, ii) also exists, it is significant only for near density-matched particles ($\gamma \approx 1$) and the sole contribution for $\gamma = 1$ (Fig. 1c, ii). Such particles do not rotate relative to the base flow, and thus cannot react to the primary base flow (Fig. 1b, i) or the rotational component of this secondary flow (Supplementary Fig. 1a, iv, 1b, iii). In such cases, a disturbance flow is generated by the extensional component of the secondary flow (Fig. 1b, ii, Supplementary Fig. 1a, v, 1b, iv). This flow 'pushes and pulls' on the particle surface, generating a viscous quadrupole disturbance flow (Fig. 1c, ii, Supplementary Fig. 2, Supplementary Movie 2). Because the secondary flow vanishes in the limit $\beta \to 0$, so does the viscous enhancement factor ($\alpha_{\rm v} \to 0$ for $\gamma = 1$, Fig. 2, $\beta \ll 1$).

For any particle that is not precisely or near density-matched ($\gamma \not\approx 1$), the viscosity-dominated rotating base flow activates the rotational inertia of the particle, converting it into a 'viscous flywheel'. While negligible inertia exists in the fluid, and there is no vertical displacement of the particle, a frequency change, $\Delta f_{\rm rot}$, occurs that depends on the particle mass density.

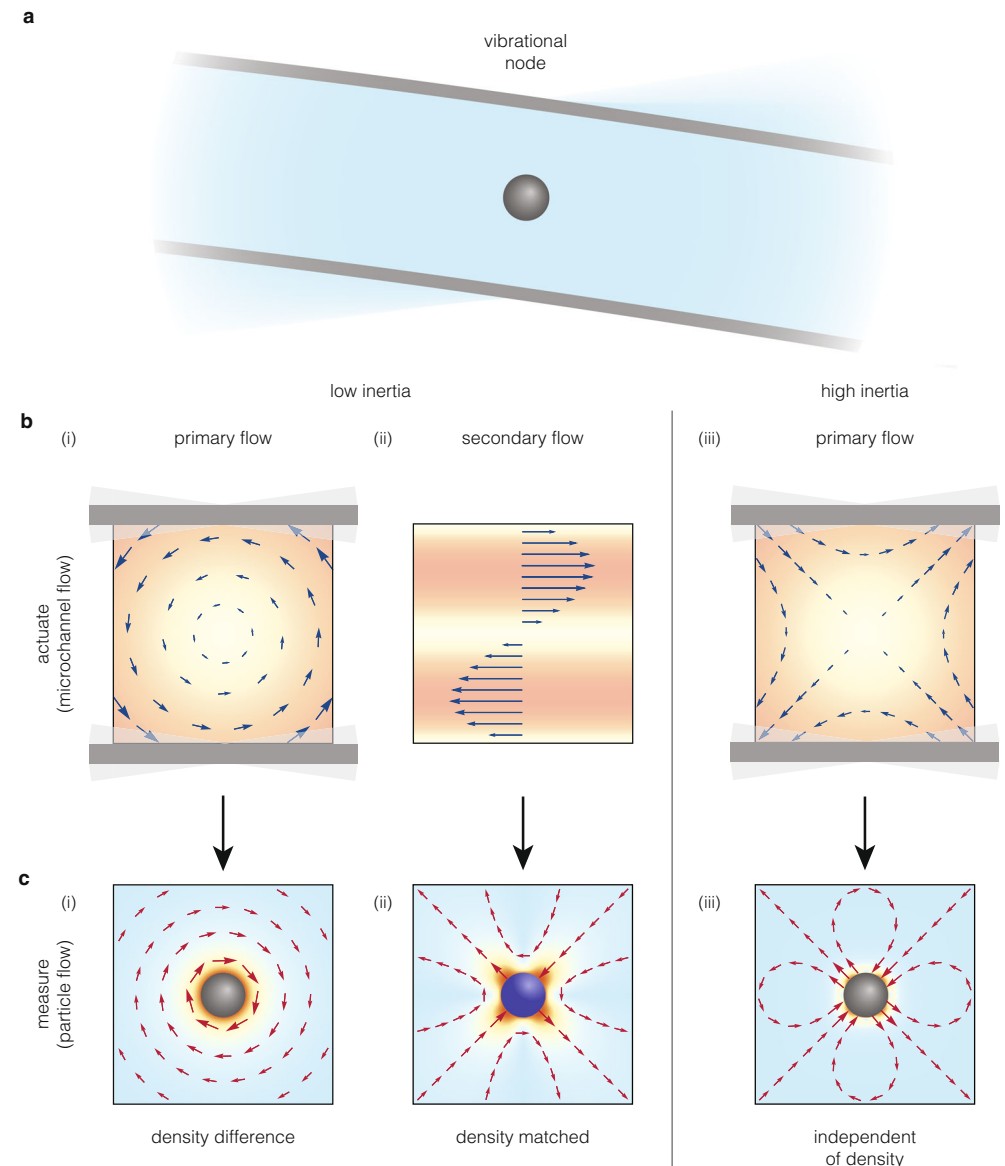

**Fig. 1 Concept of local rotation in inertial sensing. a** Schematic of a rigid particle suspended at a vibrational node of a rotating microchannel. **b** Actuating the walls with oscillatory rotation generates a flow within the microchannel (base flow). (i) At low inertia, i.e., when viscous effects dominate, the base flow is primarily a rigid-body rotation. (ii) This primary flow generates a secondary, non-linear, shear flow due to the effects of small but finite inertia. (iii) At high inertia, i.e., where viscous effects are negligible, the base flow is primarily extensional. **c** Suspending a particle in the base flow generates a disturbance flow, which produces a measurable signal based on Eq. (1). (i) At low inertia, a particle with a different density to the fluid rotates relative to the primary flow, generating a rotational disturbance flow, shown here for a negatively buoyant (heavier than fluid) particle. For a positively buoyant particle, the direction of the disturbance flow field is reversed. (ii) A particle with its density matched to the fluid does not rotate relative to the primary flow, or to the rotational component of the secondary flow. It only reacts to the extensional component of the secondary flow (Supplementary Fig. 1b, iii, iv), generating a quadrupole disturbance flow. (iii) At high inertia, the particle reacts to the extensional base flow, generating a quadrupole disturbance flow. This is in a similar manner to (ii), but independent of particle density.

For high inertia $(\beta \gg 1)$, the particle experiences an extensional base flow (Fig. 1b, iii). In contrast to $\beta \ll 1$, this base flow is inviscid, and thus irrotational, despite being driven by the rotating walls. Here, the particle does not rotate, regardless of its density, $\gamma$, and generates a quadrupole disturbance flow (Figs. 1c, iii, 2, $\beta \gg 1$, Supplementary Fig. 2, Supplementary Movie 3) similar to the density-matched particle above (Fig. 1c, ii). As a result of this irrotational base flow, the viscous enhancement factor, $\alpha_{\mathrm{v}}$, is independent of particle density, $\gamma$ (Fig. 2, $\beta \gg 1$).

In the high $\beta$ limit, the thin viscous boundary layers around the particle and in the vicinity of the walls do not overlap. Decreasing $\beta$, i.e., increasing the viscous penetration depth, $\delta$, results in a

monotonic increase to $\alpha_{\mathrm{v}}$ (Fig. 2, $\beta \gg 1$); due to an increase in the effective particle size. Ultimately, this decrease in $\beta$ causes these viscous boundary layers to overlap, producing a non-monotonic variation of $\alpha_{\mathrm{v}}$ (Fig. 2, $\beta \approx 200$). This overlap, with its resulting effect on $\alpha_{\mathrm{v}}$, is initiated at different values of $\beta$, which depend on both the particle's radius, $R$, (Supplementary Fig. 3) and its $z$-position (Supplementary Fig. 4).

**Experimental validation of theory by measuring nano- and microparticles.** To validate the theory, we experimentally measured the resonant frequency change, $\Delta f$, of six types of SMRs/SNRs cantilevers (Supplementary Table 1) while flowing

polystyrene and glass nanoparticles through their channels containing water (Supplementary Table 2). The dimensions of the cantilevers differ significantly, with effective masses, $m_{eff}$, spanning three orders-of-magnitude (Supplementary Fig. 5), while their Reynolds numbers range from $\beta \approx 30$ for the smallest

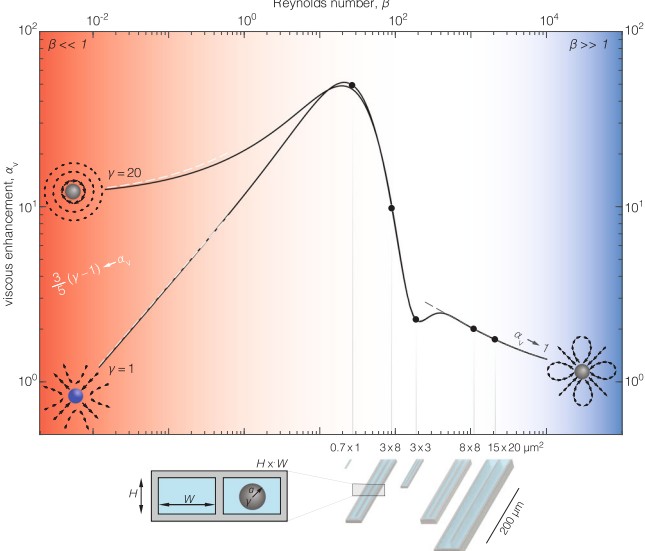

**Fig. 2 Viscous enhancement of frequency change due to local rotation.** The theoretically derived $a_v$ (black lines) expressing the viscous enhancement of signal $\Delta f_{rot}$ (Eq. (1)), is shown for $R \equiv a/H = 0.1$ where $a$ is the dimensional radius of the particle. Ratio of the particle density to the fluid density is $\gamma = \rho_p/\rho$. Similar non-monotonic behavior vs $\beta$ is observed for other $a/H$ ratios (Supplementary Fig. 3). Dashed lines represent the asymptotic limits for $\beta \ll 1$ and $\beta \gg 1$ (Supplementary Note 1). Black circles represent the cantilevers (bottom right) used in the experiments; $H \times W$ gives the dimensions of the cross-sectional area of the flow channel (bottom left). Schematics of the flow fields correspond to the disturbance flows in Fig. 1c.

device, to $\beta \approx 2,000$ for the largest (Fig. 2, black points). The smaller cantilevers exhibit increased mass responsivity while being more susceptible to the effects of fluid viscosity[19]. We actuated the cantilevers at their second flexural mode (Fig. 3a) because this mode provides the lowest frequency at which a vibrational node occurs[16]. When a particle flows through the cantilever channel, a frequency shift signal is measured containing contributions from (i) the SMR/SNR's vertical displacement, which is standard in inertial sensing[13], and (ii) its rotation (Fig. 3b). To isolate the contribution from rotation, i.e., $\Delta f_{rot}$ in Eq. (1), we developed an iterative algorithm (Supplementary Fig. 6) to process the total frequency change $\Delta f$, and extract the signal at the node $\Delta f^{node}$ (Fig. 3).

The theory predicts an inconsequential dependence of $\alpha_v$ on particle density, $\gamma$, for $\beta \gtrsim 10$ (Fig. 2, Supplementary Figs. 3, 4), indicating that $\Delta f^{node}$ in Eq. (1) depends experimentally on particle volume only. To test this prediction, we conducted two series of experiments.

First, we flowed polystyrene calibration particles with nominal radii in the range $a = 125 - 6,000$nm, and measured $\Delta f^{node}$ for each particle (Fig. 4a, Supplementary Movie 2). Next, we determined the volume, $V_{meas}$, of each particle by fitting the experimental measurements to Eq. (1) using an iterative algorithm (Supplementary Fig. 6). We also independently measured each particle's volume, $V_{ref}$—termed the reference volume—from the buoyant mass, extracted using the antinode signal[20] $\Delta f^{anti}$ (Fig. 3b). This used the known density of the particles ($\rho_{p,pol} = 1,050$kg/m$^3$) and that of the surrounding fluid ($\rho = 997$kg/m$^3$), respectively. We observed excellent agreement between $V_{meas}$ and $V_{ref}$ over the entire experimental dataset (Fig. 4b).

Experimental measurement of particle volume has three main sources of error (Supplementary Note 2), including: (i) non-linear error propagation when determining particle volume from the signal, (ii) effect of the channel walls on the disturbance flow not being accounted for by the theory, and (iii) for the devices with smaller $\beta$, error associated with assuming that the particles are at

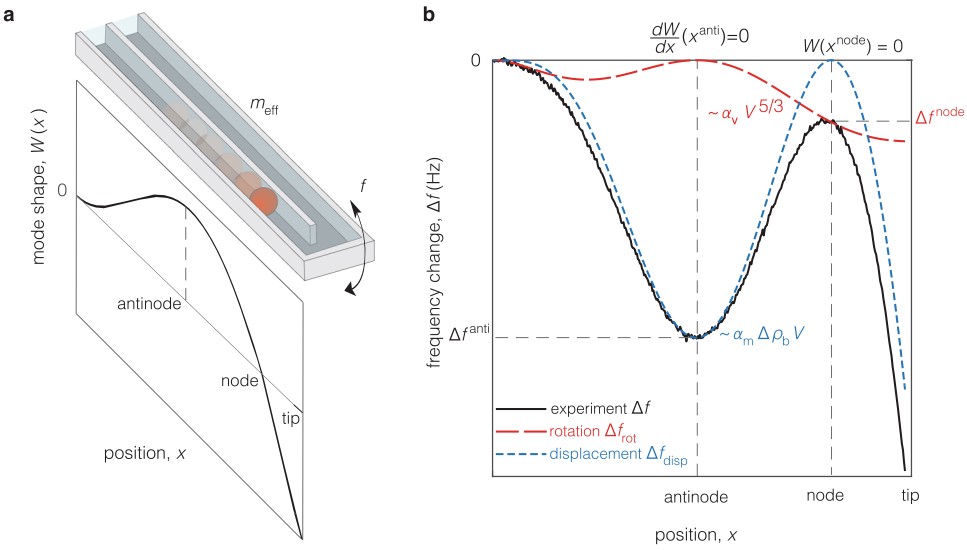

**Fig. 3 Experimental realization of local rotation in microcantilevers. a** Schematic of a particle (red) flowing inside the cantilever, with effective mass, $m_{eff}$, driven in its second flexural mode, with resonant frequency $f$; $W(x)$ specifies the displacement at a position $x$ along the length of the cantilever. At the node (i.e., where $W(x^{node}) = 0$), there is local rotation only, $dW/dx\,(x^{node}) \neq 0$. **b** Frequency change, $\Delta f$, induced by a particle with nonzero buoyant mass $m_b$ (shown here for $m_b < 0$) as the product of buoyant density, $\Delta\rho_b = \rho_p - \rho$, and volume $V$. The experimental signal, $\Delta f$ (black), consists of two signals: one due to rotation $\Delta f_{rot}$ (red), and one due to displacement $\Delta f_{disp}$ (blue). $\Delta f^{anti}$ and $\Delta f^{node}$ denote the signals at the antinode and node positions respectively. The parameter $\alpha_m$ represents the mass discrepancy in the displacement signal[22].

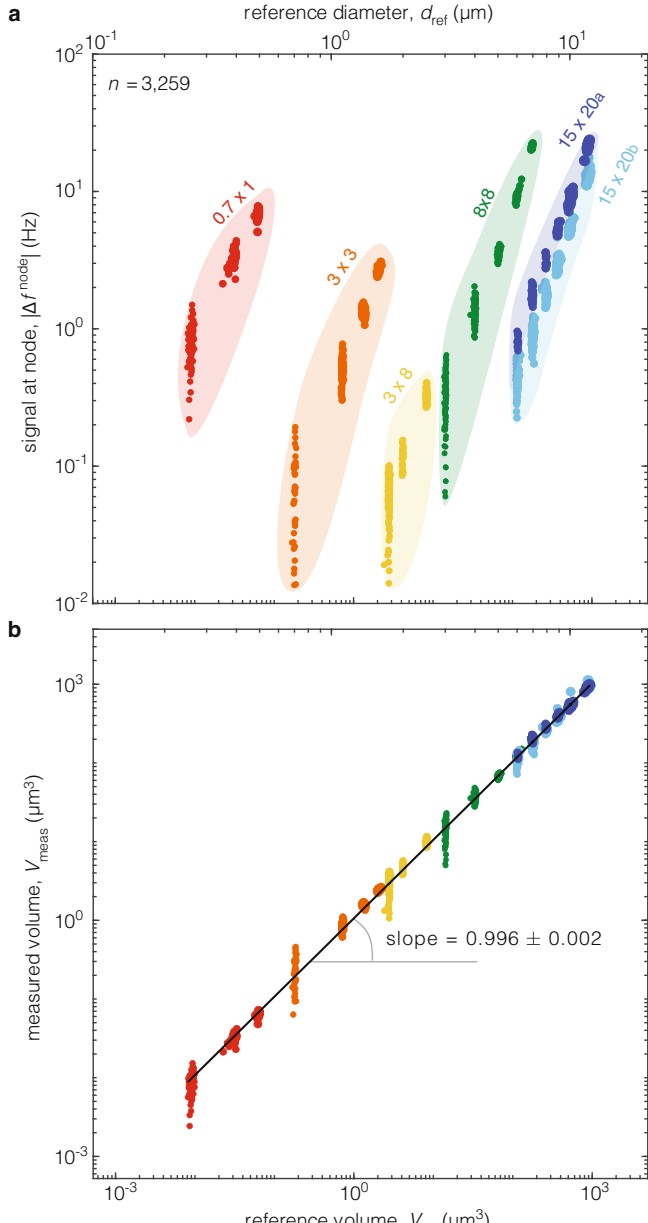

**Fig. 4 Experimental validation of theory. a** Measurements of the frequency change signal $\Delta f^{node}$ at the node position (Fig. 3b) for polystyrene particles suspended in water in six different devices. The reference size, (radius, $a_{ref}$, and volume, $V_{ref}$) is measured from the antinode signal. Note that in the present experiments, $\Delta f^{node} < 0$. The colored islands represent each different device with symbols referring to Fig. 2; $n$ is the total number of measurements across all experiments. **b** Measured volume, $V_{meas}$, vs reference particle size as in (**a**). $V_{meas}$ is calculated from the node signal, $\Delta f^{node}$, using Eq. (1) (Supplementary Fig. 6). A linear regression of $V_{meas}$ vs $V_{ref}$ is performed where the bounds give a 99.9% confidence interval.

the channel center ($z = 0$) when analyzing the experimental data; the devices with larger $\beta$ are insensitive to z-position (Supplementary Fig. 7). Even though particles may not be at $z = 0$, they maintain the same z-position as they flow through the channel due to the negligible effect of inertial migration (Supplementary Note 2)[21]. We therefore expected the distribution of z-positions to be stochastic. Using Monte-Carlo simulations (Supplementary Figs. 8, 9), signal-to-noise calculations (Supplementary Figs. 10, 11)

and scaling arguments (Supplementary Note 2), we found that all three main sources of error are minimized for $\beta \gtrsim 100$ and $R > 0.25$. Further improvement may be obtained by actuating the cantilevers at multiple modes simultaneously[20], which would enable determination of the particle z-position.

Second, we flowed glass particles of similar volume to the polystyrene particles but with a buoyant density, $\Delta\rho_p = \rho_p - \rho$, that is one order-of-magnitude larger. Analysis[22] of the measured antinode signal, $\Delta f^{anti}$, reveals an enhanced buoyant density for these glass particles of commensurately increased magnitude (Supplementary Fig. 12a, b). Even so, analysis of the measured rotation signal, $\Delta f^{node}$, directly gives a particle volume, $V_{meas}$, of similar magnitude to the polystyrene particle volumes (Supplementary Fig. 12c, d). This shows that the rotation signal at the node, $\Delta f^{node}$, is independent of particle mass and gives direct access to particle volume.

## Discussion
Our findings enable a direct measurement of particle density. Combining measurement of the particle's volume, with that of its buoyant mass, respectively using $\Delta f^{node}$ and $\Delta f^{anti}$ from a single pass of a particle through the cantilever, we measured particle density with an accuracy of at least 99% for particles that are larger than half the height of the microchannel (Supplementary Fig. 13). Previous methodologies based on fluid-filled cantilevers rely on complex fluid exchanges to measure particle mass and volume; this limits throughput to <6 particles per minute[23,24]. In comparison, the present methodology extracts these properties simultaneously from a simple, single measurement at a throughput that is 10-fold greater.

Overall, the realization of rotational inertial sensing defines a paradigm in inertial sensing for characterizing the vibrational response of fluid-suspended micro- and nanoparticles. Interaction of a rigid particle with a rotating fluid-filled microchannel leads to a rich array of flow mechanisms enabling this sensing modality. The presented analytical theory directly augments existing theory for inertial sensing that use fluid-filled cantilevers and plates[7,13]. This rigorously accounts for the ubiquitous—yet previously complicating and ignored—effects of rotation.

## Methods
**Fabrication and design of devices.** The nanochannel and microchannel suspended resonator devices (SNR, SMR) were fabricated at Innovative Micro Technology (Santa Barbara, CA, USA) and CEA-LETI (France) using 6-inch and 8-inch silicon wafer technology[8,15,16]. The technology enables the cantilevers of each device to oscillate in a dedicated vacuum cavity containing an on-chip getter to maintain the high vacuum, thus ensuring high-quality factor during operation. Each device (Supplementary Table 1) has either one cantilever (devices $0.7 \times 1.0$, $3 \times 3$) or two cantilevers (devices $3 \times 8$, $8 \times 8$, $15 \times 20a$, $15 \times 20b$). For each cantilever, there are four fluidic ports drilled on the top glass wafer to access two bypass channels respectively connected to the inlet and the outlet of each cantilever (Fig. 3a).

**Operation of devices.** Each SMR/SNR device was actuated at the second flexural mode (Fig. 3a) by a piezo-ceramic plate on top of which the device was epoxy-bonded, using a dedicated phase-locked loop (PLL) in closed loop[20]. Precision pressure regulators (electronically controlled Proportion Air QPV1 and manually controlled Omega PRG101-25) were used to flow particle solutions within each device. To measure the signal of change in resonance frequency, $\Delta f$, (Fig. 3b), either optical[25] ($0.7 \times 1$, $3 \times 3$, $3 \times 8$, $8 \times 8$) or piezoresistive readout[26,27] ($15 \times 20a,b$) methods were employed, while in both cases a field programmable gate array (FPGA, Altera Cyclone IV on DE2-115) was used, connected via ethernet cable to a desktop computer. To ensure adequate sampling[28] of $\Delta f$, the transit time $\Delta t_{transit}$ of the particle through the cantilever was generally set such that $\Delta t_{transit}(sec) > 24/bw(Hz)$, where $bw$ is the bandwidth of the PLL loop. The experiments were performed using a custom code written in LabVIEW 2017 software.

**Preparation of particle solutions.** The polystyrene calibration particles (Supplementary Table 2) were originally supplied by the vendor in aqueous solutions with concentrations of 0.20–1.00% solids. For flowing into the cantilevers, they were

diluted by a factor of 500–1000 times using purified, filtered (filter size 20 nm) water. Occasionally, to prevent the pinning of polystyrene particles inside the cantilever, Tween 20 (Sigma Aldrich, MO USA) was added at a percentage of ~0.05% per volume of particle solution. The glass particles were originally supplied by the vendor in dry form. For flowing into the cantilevers, they were suspended in purified, filtered water at concentrations of ~0.03 mg/μL with 0.0003% Tween 20. In the event of clogging, the cantilever device was flushed with a sequence of water, isopropanol, acetone, toluene until the device was unclogged.

**Post processing of experimental data**. The experimental data were analyzed by implementing an iterative algorithm (Supplementary Fig. 6) in MATLAB 2019b.

## Data availability
Source data (extracted signals at antinode and node, and rest of measured variables) are provided in the Supplementary Data. The raw experimental data (time-series data of resonant frequency) and simulation data are available from the corresponding authors upon reasonable request. Source data are provided with this paper.

## Code availability
All codes used in this study are available from the corresponding authors upon reasonable request.

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

## Acknowledgements
We thank Iris E. Hwang and Joon H. Kang for helpful comments on the manuscript, and Dimitra Vardalaki for valuable insights into designing the figures. S.R.M. and G.K. acknowledge support from the Virginia and D.K. Ludwig Fund for Cancer Research. J.F.C. and J.E.S. acknowledge support from the Australian Research Council Centre of Excellence in Exciton Science (CE170100026) and the Australian Research Council Grants Scheme. V.A. acknowledges support from LETI Carnot Institute EVEREST project for fabricating the 0.7 × 1.0 device as well as the CEA-Enhanced Eurotalents program, cofunded by FP7 Marie Skłodowska-Curie COFUND program (Grant Agreement 600382) and the Fulbright fellowship for the France National Researcher program (project title "EASY").

## Author contributions
G.K., J.F.C., J.E.S. and S.R.M. conceived the study. G.K. and S.R.M. designed the experiments. G.K. carried out the experiments with the exception of the experiments with the 3 × 3 device, which S.M.K. performed. V.A. designed and provided the 0.7 × 1.0 device. J.F.C. and J.E.S. developed the theory which built on an alternative inviscid theory developed by G.K. G.K. and J.F.C. analyzed the data and performed the Monte-Carlo simulations. G.K., J.F.C., J.E.S. and S.R.M. wrote the paper with V.A. providing input.

## Competing interests
S.R.M. is a co-founder of Travera and Affinity Biosensors, which develops technologies relevant to the research presented in this work. The rest of the authors declare no competing interests.
