## [Peer Review File · Nature Communications]

Reviewer #2 (Remarks to the Author):

This is an excellent and very original work. The extraction of an additional parameter, particle volume, from the measurement of the frequency shift in the second flexural mode will have a very high impact in the field, as this approach will allow the measurement of both particle mass and volume with very high throughput. This may have very relevant implications in the application of hollow resonators for particle classification in diverse fields, from materials science to microbiology. Furthermore, this manuscript teaches also new physics of hollow resonators. It presents a detailed theoretical analysis of flow disturbances due to particles in liquid-filled resonators and the effect on the vibration, with an excellent analysis of the noise sources and elegant and useful analytical expressions to interpret experimental measurements. The proposed iterative algorithm is practical and will be of high interest to the field. The experimental validation is thorough as six cantilever types and different particle materials and sizes have been measured so I am fully convinced of the validity of this new approach.

We thank the reviewer for the positive feedback.

The authors briefly discuss the error due to particle position in the z coordinate in supplementary note 2 but I think the readers would benefit of a more in-depth discussion about this. The mid-height position may not be the most probable due to lift and drag forces (Lab Chip, 2016,16, 10-34, Scientific Reports 9, 2019, 16575). Can the authors calculate or measure the experimental z position of the particles in their channels? Are they truly travelling close to the channel center position? The case of particles travelling in contact with the channels' wall might be particularly interesting and modelling of this scenario is at reach. This particular case is commonly found experimentally and it has been observed experimentally also with transparent hollow resonators (ACS Sens 2019, 4, 12, 3325), so it will be of interest to readers in the field.

An important finding of our study is that the larger devices, which are most commonly used in practice, are insensitive to z-position. Only the small devices exhibit sensitivity to z-position. We have now clarified this important distinction in the manuscript (lines 217-224) and expanded our discussion on the z-position error (Supplementary Note 2, new text highlighted in gray). Furthermore, we have completely reanalysed all experimental data adding a new figure S13 to quantify this error. We found appreciable error is only present for the smallest particles in the smallest devices ($R < 0.25$ and $\beta < 100$).

In the current configuration we are not able to measure the z-position of the particles and we agree that they may not always flow down the centerline of the channel. However – for the reasons outlined above – this does not compromise the overall accuracy of our approach.

To summarise the new text in Supplementary Note 2 and lines 220-224: The particles randomly enter the channel and initially exhibit a stochastic distribution. A scaling argument reveals that lift forces in the channel are negligible and thus particles follow a single z-position as they flow through the SMR. Although it is possible for some particles to roll along the walls, this will be a negligible contribution to the overall experiments that consists of hundreds of individual measurements.

Reviewer #4 (Remarks to the Author):

The paper by Manalis and Sader is, as always, a fantastic piece of research, incredibly impressive. Very well cared and written. Figures are excellent and experimental data are astounding.

We thank the reviewer for the positive feedback.

I think that the paper should be accepted but I would like to ask the authors a couple of questions and ask for a couple of modifications:

- Figure 2 has 2 typos - enChancement and α_v instead of α_v

We have corrected the typo in the revised manuscript.

- Also in Fig 2 - why that little dip at $\beta=200$? I did not get if this was predicted or if it is some sort of extrapolation to fit the asymptotes.

This small decrease or "dip" in the viscous enhancement parameter α_v is predicted by the theory (see lines 168-176). We have not performed any fits to the asymptotes, they only serve to provide simple formulae to the full theory in these limits.

- Before Eq. 1, you should already refer to Supp note 1. it is there that you include the Δf_a and Δf_n . and you should also mention somehow in the text of that paragraph that "in addition to the effect of the buoyant mass"... otherwise it seems as if you are forgetting the effect that you have been measuring for 15 years (oh gosh! it's scary how time goes by).

To improve the clarity of our manuscript, we amended the text at lines 106-109 of revised manuscript, explicitly mentioning the standard buoyant mass measurements.

- then I was wondering about the explanation you give: "The disturbance flow alters the torque exerted by the fluid on the channel walls. This change in torque modifies the effective rigidity of the cantilever, shifting its resonant frequency"

Can you elaborate a bit on this? to make it more understandable without going to your supp note 1 and having to understand that derivation? I mean, why the torque exerted by the fluid on the walls is affecting the frequency? is it stress? is it channel dimensions? because I am guessing (only guessing now) that it is the torque that changes the frequency and not the shift in the torque, right? or am I wrong?

We accordingly have revised the text (see lines 102-105) to explain what causes the shift in the resonant frequency.

- what happens if the channel is NOT centered with the neutral axis???)

We have added new text (line 68-72 and Supplementary Note 2) to discuss off-axis placement of the channel; it does not affect the experimental results in this study but is an intriguing area for a future investigation.

- In fig 3 - why don't you put in the legend Δf_n and Δf_a ? or maybe I am not understanding something. I mean, this is a great point you are making in this figure, where you very clearly show that: the old effect is zero at the node, but the new effect still exists there. whereas in the antinode, the new effect is zero.

In the original manuscript, we had not included Δf_n and Δf_a in the legend of Fig 3 because they refer to frequency changes at specific positions of cantilever (node and antinode) rather than whole curves of frequency change due to rotation and displacement respectively. To clarify this point, we have now changed the symbols in the revised manuscript referring to:

Δf_{rot} : frequency change due to local rotation at an arbitrary position of the resonator. This is in equation 1 and red curve in Fig. 3b (incorporated now in legend)

Δf_{disp} : frequency change due to local displacement at an arbitrary position of the resonator (standard theory). This is the blue curve in Fig. 3b (incorporated now in legend).

Δf^{node} , Δf^{anti} : frequency changes at node and antinode positions of the resonator (same as before).

- Last point: 2 years ago or so the authors published a paper where they were measuring the stiffness of the particles flowing in the channel. where is that effect here? why am I not seeing it in the equations?

While in the originally submitted manuscript we had stated the work focused on rigid particles, we agree that this point could have been easily missed. We have revised the manuscript to make this clearer by adding the word 'rigid' to the abstract and conclusions. While in our previous work we were able to measure stiffness of non-rigid particles from the node signal, we required particle volume to be known *a priori*. Here, we independently measured particle volume for rigid particles based on our new analytical theory and experimental validation.

REVIEWER COMMENTS

Reviewer #2 (Remarks to the Author):

This is an excellent and very original work. The extraction of an additional parameter, particle volume, from the measurement of the frequency shift in the second flexural mode will have a very high impact in the field, as this approach will allow the measurement of both particle mass and volume with very high throughput. This may have very relevant implications in the application of hollow resonators for particle classification in diverse fields, from materials science to microbiology. Furthermore, this manuscript teaches also new physics of hollow resonators. It presents a detailed theoretical analysis of flow disturbances due to particles in liquid-filled resonators and the effect on the vibration, with an excellent analysis of the noise sources and elegant and useful analytical expressions to interpret experimental measurements. The proposed iterative algorithm is practical and will be of high interest to the field. The experimental validation is thorough as six cantilever types and different particle materials and sizes have been measured so I am fully convinced of the validity of this new approach.

The authors briefly discuss the error due to particle position in the z coordinate in supplementary note 2 but I think the readers would benefit of a more in-depth discussion about this. The mid-height position may not be the most probable due to lift and drag forces (Lab Chip, 2016,16, 10-34, Scientific Reports 9, 2019, 16575). Can the authors calculate or measure the experimental z position of the particles in their channels? Are they truly travelling close to the channel center position? The case of particles travelling in contact with the channels' wall might be particularly interesting and modelling of this scenario is at reach. This particular case is commonly found experimentally and it has been observed experimentally also with transparent hollow resonators (ACS Sens 2019, 4, 12, 3325), so it will be of interest to readers in the field.

Reviewer #4 (Remarks to the Author):

The paper by Manalis and Sader is, as always, a fantastic piece of research, incredibly impressive. Very well cared and written. Figures are excellent and experimental data are astounding.

I think that the paper should be accepted but I would like to ask the authors a couple of questions and ask for a couple of modifications:

- Figure 2 has 2 typos - enChancement and a_v instead of α_v
- Also in Fig 2 - why that little dip at $\beta=200$? I did not get if this was predicted or if it is some sort of extrapolation to fit the asymptotes.
- Before Eq. 1, you should already refer to Supp note 1. it is there that you include the Δf_a and Δf_n . and you should also mention somehow in the text of that paragraph that "in addition to the effect of the buoyant mass"... otherwise it seems as if you are forgetting the effect that you have been measuring for 15 years (oh gosh! it's scary how time goes by).
- then I was wondering about the explanation you give: "The disturbance flow alters the torque exerted by the fluid on the channel walls. This change in torque modifies the effective rigidity of the cantilever, shifting its resonant frequency"

Can you elaborate a bit on this? to make it more understandable without going to your supp note 1

and having to understand that derivation? I mean, why the torque exerted by the fluid on the walls is affecting the frequency? is it stress? is it channel dimensions? because I am guessing (only guessing now) that it is the torque that changes the frequency and not the shift in the torque, right? or am I wrong?

- what happens if the channel is NOT centered with the neutral axis??? :)

- In fig 3 - why don't you put in the legend Δ_n and Δ_a ? or maybe I am not understanding something. I mean, this is a great point you are making in this figure, where you very clearly show that: the old effect is zero at the node, but the new effect still exists there. whereas in the antinode, the new effect is zero.

- Last point: 2 years ago or so the authors published a paper where they were measuring the stiffness of the particles flowing in the channel. where is that effect here? why am I not seeing it in the equations?

REVIEWERS' COMMENTS

Reviewer #2 (Remarks to the Author):

The manuscript is suitable for publication. Great work!!!

Reviewer #4 (Remarks to the Author):

Thanks for answering all my queries.
The paper looks great.